# Unveiling Fungi Armor: Preliminary Study on Fortifying *Pisum sativum* L. Seeds against Drought with *Schizophyllum commune* Fries 1815 Polysaccharide Fractions

**DOI:** 10.3390/microorganisms12061107

**Published:** 2024-05-29

**Authors:** Jovana Mišković, Gordana Tamindžić, Milena Rašeta, Maja Ignjatov, Nenad Krsmanović, Gordana Gojgić-Cvijović, Maja Karaman

**Affiliations:** 1ProFungi Laboratory, Department of Biology and Ecology, Faculty of Sciences, University of Novi Sad, Trg Dositeja Obradovića 2, 21000 Novi Sad, Serbia; milena.raseta@dh.uns.ac.rs (M.R.); nenad.krsmanovic@dbe.uns.ac.rs (N.K.); 2Institute of Field and Vegetable Crops, National Institute of the Republic of Serbia, 21000 Novi Sad, Serbia; gordana.tamindzic@ifvcns.ns.ac.rs (G.T.); maja.ignjatov@ifvcns.ns.ac.rs (M.I.); 3Department of Chemistry, Biochemistry and Environmental Protection, Faculty of Sciences, University of Novi Sad, Trg Dositeja Obradovića 3, 21000 Novi Sad, Serbia; 4Department of Chemistry, Institute of Chemistry, Technology and Metallurgy, University of Belgrade, Njegoševa 12, 11000 Belgrade, Serbia; ggojgic@chem.bg.ac.rs

**Keywords:** agricultural application, fungi, biopriming, stress condition, submerged cultivation, *S. commune*, *P. sativum*

## Abstract

Amidst worsening climate change, drought stress imperils global agriculture, jeopardizing crop yields and food security, thereby necessitating the urgent exploration of sustainable methods like biopriming for the harnessing of beneficial microorganisms to bolster plant resilience. Recent research has revealed diverse biological compounds with versatile applications produced by *Schizophyllum commune*, rendering this fungus as a promising contender for biopriming applications. For the first time, this study aimed to investigate the potential of *S. commune* exo- (EPSH) and intra-polysaccharides (IPSH) isolated from two strains—Italian (ITA) and Serbian (SRB)—under submerged cultivation to enhance the resilience of *Pisum sativum* L. seeds through the biopriming technique. Testing of the seed quality for the bioprimed, hydroprimed, and unprimed seeds was conducted using a germination test, under optimal and drought conditions, while characterization of the PSHs included FTIR analysis, microanalysis, and determination of total protein content (TPC). The FTIR spectra of EPSH and IPSH were very similar but revealed the impurities, while microanalysis and TPC confirmed a different presence of proteins in the isolated PSHs. In optimal conditions, the IPSH SRB increased germination energy by 5.50% compared to the control; however, the highest percentage of germination (94.70%) was shown after biopriming with the PSH isolated from the ITA strain. Additionally, all assessed treatments resulted in a boost in seedling growth and biomass accumulation, where the ITA strain demonstrated greater effectiveness in optimal conditions, while the SRB strain showed superiority in drought conditions. The drought tolerance indices increased significantly in response to all examined treatments during the drought, with EPSH ITA (23.00%) and EPSH SRB (24.00%) demonstrating the greatest effects. Results of this preliminary study demonstrate the positive effect of isolated PSH, indicating their potential as biopriming agents and offering insights into novel strategies for agricultural resilience.

## 1. Introduction

In the face of escalating climate change, drought stress poses a significant threat to global agriculture, compromising crop productivity and food security [1]. Finding sustainable and efficient methods to bolster plant resilience against drought stress has become one of objectives in agricultural research. Among the various strategies to mitigate drought stress, biopriming—an eco-friendly technique involving the use of beneficial microorganisms—has emerged as a promising approach to enhance crop resilience [2].

Biopriming is a cutting-edge seed treatment technique that involves the application of beneficial microorganisms, such as fungi, bacteria, or actinomycetes or their products onto the seed surface to enhance seed germination, seedling vigor, and overall plant health [3]. This innovative method aims to improve seed quality, uniformity, and establishment while mitigating the adverse effects of various biotic and abiotic stresses on plants. Biopriming with these beneficial microorganisms has shown promising results in enhancing plant growth, promoting antioxidative defense systems, and increasing biomass and yield in various crops [4]. Moreover, it helps plants cope with abiotic stresses like drought, salinity, or low temperatures by improving water uptake and enhancing root and shoot growth [5]. Also, this technique is considered an eco-friendly alternative to chemical treatments, contributing to sustainable agriculture practices [6].

Pea *(Pisum sativum* L.) is a plant species that has long served as a model organism for understanding various aspects of plant biology and is cultivated globally for its nutritional value and versatility in culinary applications [7]. Consumption of *P. sativum* L. has been associated with various health benefits beyond basic nutrition, since the plant’s components, including starch, protein, fiber, vitamins, minerals, and phytochemicals, offer antioxidant properties, gastrointestinal health benefits, reduced glycemic index due to its intermediate amylose content, and potential bioactivities like angiotensin I-converting enzyme inhibition and antioxidant activity [8]. Hence, vegetable peas in the last 20 years have gained significant attention all over the world [9]. Moreover, the pea plays a vital role in agriculture as a legume crop that provides protein-rich feed and contributes to sustainable farming practices through nitrogen fixation [10]. Its cultivation supports soil health by enhancing nitrogen levels through symbiotic relationships with nitrogen-fixing bacteria [10]. However, pea susceptibility to drought stress, underscores the necessity to reinforce its defenses against such environmental challenges [11].

*Schizophyllum commune* Fries (1815) stands out for its adaptability, while recent studies have unveiled the different biological compounds synthesized by *S. commune* as potent bioactive compounds with multifaceted applications [12,13,14]. Research has shown that this fungus produces compounds like alkaloids, flavonoids, phenols, saponins, and tannins that can affect other microorganisms [15]. Moreover, over the years, several studies have highlighted the polysaccharides (PSH) of *S. commune* with immunomodulatory effects on humans, antioxidant, and antimicrobial properties against *Escherichia coli*, *Bacillus cereus*, *Staphylococcus aureus* and *Salmonella* sp. [16,17]. The most investigated PSH from *S. commune* is a beta-glucan, schizophyllan, that has been found to exhibit diverse biological effects, including antitumor, immunomodulatory, and anti-inflammatory activities [18]. In our previous research, we identified PSHs isolated from the Italian (ITA) and Serbian (SRB) strains of *S. commune* after submerged cultivation as a β-glucan complex [17] and proved its anti-acetylcholinesterase and antioxidant properties, which makes *S. commune* a promising candidate for a biopriming agent that can positively influence seed germination and plant growth and development. However, the potential of these PSHs in providing drought resistance to agricultural crops has remained unexplored.

This preliminary research aims to unveil the hidden armor that fungi and their biological compounds like PSHs provide to plants, shedding light on a novel approach to enhance crop survival under adverse environmental conditions, since to the best of our knowledge, macrofungi have never been examined in this context before. Therefore, in this research, we have examined and contrasted, for the first time, the effect of PSH derived from submerged cultivation of two *S. commune* strains originating from Italy and Serbia, on the seed quality performance of *P. sativum* L. subsequent to biopriming under both, optimal and drought conditions.

## 2. Materials and Methods

### 2.1. Fungal Material

Two dikaryon strains of the wild-growing *S. commune* Fries (1815), belonging to the Phylum Basidiomycota, Class Agaricomycetes, Order Auriculariales, and Family Schizopyllaceae, were collected in 2016 near Bologna (Italy, IT) and in 2012 in Zmajevac (Fruška Gora low Mountain chain) in Serbia (SRB). Identification of the fungal species was conducted through a study of their fungal morphology both macroscopically (considering color, shape, size, and hyphae) and microscopically. Mycelia were extracted from the fruiting bodies of both isolates and then cultured at 26 °C for 10 days on malt agar (Torlak, Serbia). These mycelia from both isolates were preserved in the fungal culture collection FUNGICULT at the ProFungi laboratory (Department of Biology and Ecology, Faculty of Sciences, University of Novi Sad; https://www.pmf.uns.ac.rs/en/research/groups/profungi/, accessed on 14 April 2024). Each isolate was assigned a specific reference number as follows: 0043 for *S. commune* SRB and 0047 for *S. commune* IT as presented in Mišković et al. [17].

### 2.2. Fungal Cultivation and Polysaccharide Extraction Process

The polysaccharide (PSH) extracts were prepared following a modified method outlined by Chen et al. [19]. PSH extracts from both the *S. commune* strains (SRB and ITA) were simultaneously extracted to obtain the exo-polysaccharide (EPSH) from the filtrate (F) and the intra-polysaccharide (IPSH) from the mycelia biomass (BM). The extraction process began with inoculating the respective fungal strains from the culture collection (FUNGICULT, ProFungi laboratory, Faculty of Sciences, University of Novi Sad) onto malt agar (Torlak, Belgrade, Serbia) and cultivating them in a thermostat (IKA-Werke GmbH & Co. KG, Staufen in Breisgau, Germany) at 26 °C for 12 days. Subsequently, plaques (1 cm^2^) were transferred into 100 mL of liquid medium [17] for submerged cultivation on a thermostatic shaker (120 rpm, 26 °C; IKA, KS 4000, IKA, Staufen, Germany) for 14 days. After cultivation, the samples were filtered to separate BM from the F components. The EPSH was then precipitated overnight at 4 °C using absolute ethanol (Sigma-Aldrich (Steinheim, Germany), followed by filtration and centrifugation twice at 4 °C (10,000× *g*, 20 min; Centrifuge 5810 R, Eppendorf, Hamburg, Germany) to remove the supernatant. The resulting precipitate was dried at +60 °C (Memmert UF55, Memmert, Büchenbach, Germany) for 20 min to eliminate residual water and ethanol and subsequently rehydrated in distilled water (dH_2_O) on a magnetic stirrer (Velp Scientifica, Usmate Velate, Italy) at 85 °C, 100 rpm, for one hour.

BM underwent freezing, lyophilization (Christ Alpha 2-4 LD plus, Martin Christ, Osterode am Harz, Germany) and grinding (IKA A11, Staufen, Germany). Subsequently, the BM was hydrated (0.1 g in 10 mL dH_2_O) and dried at 121 °C for 20 min (Memmert UF55 oven, Memmert, Büchenbach, Germany). This hydration and drying process was repeated three times prior to centrifugation at 8000 rpm for 10 min (Centrifuge 5810 R, Eppendorf, Hamburg, Germany). Each drying cycle was followed by the addition of dH_2_O. The resulting supernatant contained the extracted intra-PSH (IPSH).

### 2.3. Characterization of Isolated EPSH and IPSH

#### 2.3.1. FTIR Analysis

The FTIR spectra of the isolated EPSH and IPSH were analyzed using a Thermo-Nicolet Model 6700 spectrophotometer (Thermo Fisher Scientific, Madison, WI, USA) fitted with a Smart Orbit (Diamond) ATR accessory (Thermo Fisher Scientific, Waltham, MA, USA) and OMNIC 7.3 software.

#### 2.3.2. Elemental Organic Microanalysis

Elemental organic microanalysis was conducted using a Vario EL III CHNS/O elemental analyzer from Elementar (Hanau, Germany) to determine carbon, hydrogen, nitrogen, and sulfur contents. Prior to microanalysis, the samples underwent preparation involving drying at +105 °C until a constant mass was achieved.

#### 2.3.3. Quantification of Total Protein Content (TPC)

The protein content in isolated ISPH from both strains was assessed using the dye-binding colorimetric method [20]. Absorbance readings were taken at a wavelength of 595 nm after 5 min. Protein concentration was determined from the calibration curve of bovine serum albumin (BSA) in three repetitions, and the mean value (mg BSA/g d.w.) was calculated.

### 2.4. Plant Material and Seed Priming

The plant material utilized in this study was the garden pea (*Pisum sativum* L.) cv. Dunav, developed at the Institute of Field and Vegetable Crops, National Institute of the Republic of Serbia, Novi Sad, Serbia, within the Department of Vegetable and Alternative Crops. The seeds of the chosen pea cultivar were produced at the Rimski Šančevi (45°19′ N, 19°50′ E), Serbia, in 2022.

For seed priming, the pea seeds underwent sterilization with 5% sodium hypochlorite solution (NaClO) (Sigma Aldrich, St. Louis, MO, USA) followed by a triple rinsing with distilled water. Seed priming was carried out by immersing the pea seeds in dH_2_O (hydropriming—HP) and aqueous solutions containing PSH extracts at a concentration of 1% (10 mg/mL) (both EPSH and IPSH) in a ratio of 1:5 (*w*/*v*) as per Farooq et al. [21] for 6 h following Arafa et al. [22]. Biopriming involved immersion in EPSH and IPSH solutions from both the ITA and SRB fungal strains, while the control group remained unprimed. Subsequently, the seeds were rinsed thoroughly with dH_2_O and air-dried on filter paper until they regained their initial weight.

### 2.5. Examination of Seed Quality and Initial Growth of Pea Plants under Optimal and Drought Conditions

#### 2.5.1. Seed Germination Assessment

The study involved a working sample comprising 3 sets of 100 randomly selected seeds each. Along with the control group, these seeds were placed post priming in plastic boxes sized 240 × 150 mm, with sterilized sand serving as the growth medium. The experiment consisted of a total of 36 boxes, grouped into two sets, where one set contained 18 boxes that were optimally supplied with water, and the other contained 18 boxes in which a drought was simulated using a −0.5 MPa solution of polyethylene glycol (PEG-6000) (Sigma Aldrich, St. Louis, MO, USA), which had been proven to be a drought threshold (medium stress) that significantly reduces pea seedling growth, as outlined by Tamindžić et al. [23]. Both sets were placed in a germination chamber (Conviron CMP4030, Winnipeg, MB, Canada) at +20 °C for eight days following the ISTA Rules [24]. Germination energy (GE) (first count) was assessed on the fifth day post sowing by counting only normal seedlings with well-developed essential structures, while seed germination (SG) and the percentage of abnormal seedlings (AS) were evaluated on the eighth day post sowing.

#### 2.5.2. Determination of the Shoot and Root Length and the Root/Shoot Ratio

The determination of the shoot length (SL) and the root length (RL), as well as the root/shoot ratio (R/S ratio), was conducted by placing 25 seeds per replicate on filter paper moistened with water (optimal conditions) and a −0.5 MPa PEG solution (simulating water deficit—drought). This setup was then incubated in a germination chamber (Conviron CMP4030, Canada) at 20 °C for eight days. The shoot and root length of 10 normal seedlings were measured on the fifth and eighth day using a ruler [25]. The Root/Shoot ratio (R/S ratio) was calculated on the eighth day based on a formula provided by Bayat et al. as follows [26]:Root/Shoot ratio = Average root length (mm)/Average shoot length (mm)

#### 2.5.3. Determination of the Fresh and Dry Shoot and Root Biomass Accumulation

On the eighth day following seed placement on filter paper, the shoot fresh weight (SFW) and root fresh weight (RFW) were measured using an analytical balance (Kern 770-13, KERN & Sohn GmbH, Ballingen, Germany). Subsequently, the samples underwent drying in an oven for 24 h at 80 °C (Heraeus UT 12 Oven, Heraeus Instruments, Hanau, Germany), after which the shoot dry weight (SDW) and root dry weight (RDW) of the pea seedlings was determined.

#### 2.5.4. Determination of Shoot Elongation Rate (SER) and Root Elongation Rate (RER)

The shoot elongation rate (SER) and root elongation rate (RER) were calculated following the method of Channaoui et al. [27] in three replicates. The formulas used to determine these rates were based on the methodology outlined in the provided research:SER = (SLE − SLS)/(TE − TS)
RER = (RLE − RLS)/(TE − TS)
where SLE represents the shoot length, and RLE the root length determined on the fifth day, while SLS and RLS show the shoot and root length of the seedlings determined on the eighth day, while TE and TS are the periods (days) between the two measurements.

#### 2.5.5. Determination of Seed Vigor Index

Seedling Vigor Index (SVI) [28] was determined based on the following formula:SVI = SL × FG
where SL is seedling length (cm), and FG represents final germination (%).

#### 2.5.6. Electrolyte Permeability Assay

The impact on membrane permeability was evaluated by assessing electrolyte leakage (EL) following the protocol outlined by Blum and Ebercon [29] as detailed by Farooq et al. [21]. Six leaf discs from the pea plants were washed with distilled water, soaked in 6 mL of distilled water for 12 h. The electrical conductivity of the solution (S1) was then measured using a laboratory conductometer (HI5321, Hanna Instruments, Woonsocket, RI, USA). Subsequently, the samples were subjected to boiling water for 20 min, cooled to room temperature, and the electrical conductivity of the solution (S2) was recorded. Electrolyte permeability (EL) is calculated as the ratio of S1 to S2.

#### 2.5.7. Determination of Membrane Stability Index

The membrane stability index (MSI), as defined by Sairam [30] and elaborated by Tamindžić et al. [25], was determined through a procedure involving two sets of test tubes. Each set contained 0.10 g of fresh leaf mass and 10 mL of distilled water. One set was subjected to heating at 30 °C for 30 min using a water bath (VIMS elektrik, WKP-14, Tršić, Serbia), followed by measurement of electrical conductivity (S1) using a laboratory conductometer (Laboratory Research Grade Benchtop EC/TDS/Salinity/Resistivity Meter—HI5321, Hanna Instruments, Woonsocket, RI, USA). The second set underwent heating at 100 °C for 15 min, after which the electrical conductivity (S2) was determined. The MSI was then calculated using the provided formula:MSI = (1 − C1/C2) × 100

#### 2.5.8. Determination of Relative Water Content

The relative water content (RWC) in pea leaves was determined using the procedure outlined in Farooq et al. [21]. For this test, approximately 0.50 g of fresh leaf mass (Wf) was measured, then the leaves were rinsed and immersed in tubes filled with water until fully saturated, followed by another measurement (Ws). The saturated leaves were then dried in an oven for 24 h (at 80 °C), and their mass was measured again (Wd). The relative water content was subsequently calculated using the following formula:RWC = (Wf − Wd)/(Ws − Wd) × 100

#### 2.5.9. Determination of Pea Stress Tolerance Indices

The drought tolerance index (DTI), i.e., plant tolerance to water deficit or stressful conditions (drought), recalculated according to Maiti et al. [31]:The drought tolerance index (DTI) = Dry plant biomass in control group (g)/Dry plant biomass in treated group (g)

The shoot length stress tolerance index (SLSI) and root length stress tolerance index (RLSI) were determined using the following formulas [26]:SLSI = Average shoot length in treatment (mm)/Average shoot length in control (mm)
RLSI = Average root length in treatment (mm)/Average root length in control (mm)

### 2.6. Statistical Analysis

The obtained data were processed statistically using analysis of variance (One-way and Two-way ANOVA), while the significance of the differences between means was tested using Tukey’s HSD test at the significance level of *p* < 0.05. Correlation analysis was performed using Pearson’s product–moment correlation. The aforementioned statistical analyses were performed in IBM SPSS statistical software (version 22.0 for Windows) and Statistica software version 12.01 (StatSoft Inc., Tulsa, OK, USA), while the heat map was performed in Microsoft Excel (version 2016).

## 3. Results

### 3.1. FTIR Analysis, Microanalysis, and Quantification of Total Protein Content (TPC)

The FTIR spectra of the analyzed EPSH and IPSH were very similar and showed the co-presence of impurities, i.e., aromatic compounds, and proteins (Figure 1). This assertion was corroborated by microanalysis, which confirmed the existence of nitrogen in the IPSHs across both strains, suggesting the presence of proteins or other nitrogen-containing polymers, presumably chitin (Table 1). Notably, the nitrogen content in the IPSH varied between the ITA (3.15%) and SRB (2.75%) strains, with the percentages of carbon and hydrogen remaining relatively consistent (Table 1), while total protein content (TPC) revealed levels below 1% in both strains (ITA: 0.6%, SRB: 0.9%) (Table 1).

### 3.2. Effect of Seed Biopriming Treatments on Seed Germination and Initial Seedling Growth of Pea

The two-factor analysis of variance (Table 2) showed that the stress factor (S) had a significant effect on all examined parameters at a significance level of *p* < 0.001, except for the AS parameter (*p* < 0.01). Also, the treatment factor (T) had a statistically significant effect on all the examined parameters at the significance level of *p* < 0.001, except for the AS and SFW parameters where the significance was manifested at the *p* < 0.01 level. In addition, the interaction of S × T had a statistically significant effect on all examined parameters (*p* < 0.001), except for the AS parameter, which did not show significance, while the RWC parameter showed significance at the level of *p* < 0.01.

Examining the impact of biopriming with the PSHs derived from *S. commune* under distinct growth conditions (optimal conditions, drought stress) revealed the effect on enhancing the peas’ GE and SG under optimal conditions. Specifically, the IPSH s SRB extract demonstrated a 5.50% increase in GE compared to the control, with the highest SG percentage observed in seeds treated with PSH (EPSH and IPSH) isolated from the ITA strain (94.70%) (Figure 2). However, no significant differences were observed in the occurrence of AS among the treatments investigated. Conversely, drought stress induced by the PEG solution significantly reduced the GE of the pea seeds by 34.50% in the control group, with a 12.60% reduction in SG under drought conditions. Nonetheless, analysis of variance highlighted a positive impact of PSH on GE and SG compared to control and HP (*p* < 0.05). Particularly, biopriming with the IPSH ITA extract led to a 46.00% increase in GE and a 16.00% increase in SG under drought conditions (Figure 2 and Figure 3). Seeds treated with PSH from the SRB strain exhibited 31.60% (EPSH SRB) and 35.20% (IPSH SRB) higher GE and 9.70% and 13.90% higher SG compared to the control group (Figure 2). Notably, the occurrence of AS decreased under both optimal and drought conditions (Figure 2 and Figure 3).

The quality parameters of initial pea growth following biopriming with fungal PSH were examined under optimal conditions and drought stress as well. Biopriming treatments significantly affected pea growth parameters in both conditions, as indicated in Table 3 and Table 4.

In optimal conditions, the pea SL averaged 55 mm in the control group. However, all tested treatments exhibited a statistically significant increase in SL compared to the control (*p* < 0.05). The most considerable increase was observed with biopriming using the ITA strain extracts (17.30% and 10.60%, respectively). Drought stress resulted in a 52.70% reduction in SL, with all investigated treatments showing a positive effect on this parameter. Nevertheless, under drought conditions, the SRB strain extracts demonstrated a superior effect compared to other treatments, leading to a 45% and 43.50% increase in SL, respectively, compared to the control. Additionally, biopriming significantly impacted shoot RL under both optimal and drought conditions compared to control and HP. Notably, the highest values of RL were recorded with biopriming using the SRB strain extracts (Table 3). Biopriming of seeds under optimal conditions did not have a statistically significant effect on the SFW compared to the control. However, in conditions of water deficit, treatments of the SRB strain with PSH had a significant effect on this parameter since it increased by 37.30% after biopriming with EPSH SRB and by 20.30% after treatment with IPSH SRB, compared to the control. When it comes to the RFW, it can be clearly observed that the drought stress had a negative effect on this parameter, where the analysis of the results determined that the reduction is 53.00% compared to the control (Table 3). Also, analysis of variance showed a significant effect of treatment on this parameter both in optimal conditions and in drought stress conditions. All treatments led to an increase in the fresh mass of roots under optimal conditions; only the treatment with the IPSH SRB extracts was not statistically significant. The highest values of this parameter were recorded after biopriming with the EPSH ITA and IPSH SRB extracts (17.80% and 13.90%, respectively) compared to the control. In conditions of water deficit, analysis of variance showed that all treatments had a positive and statistically significant effect on this parameter. Also, a similar pattern of biopriming influence was observed under optimal conditions, where EPSH ITA and IPSH SRB extracts exhibited the best effect on RFW compared to the control (15.50% and 27.50%, respectively) (Table 3).

Similar to previous parameters, drought led to a reduction in SDW by 44.80% and a reduction in RDW by 40.10% compared to the control. Under optimal conditions, the only significant increase in SDW was observed after treatment with the EPSH ITA extracts (24.50%), compared to the control. In contrast, under stressful drought conditions, all the examined treatments significantly increased the SDW compared to the control. The highest value of SDW was recorded following biopriming with the EPSH SRB extracts (0.1434 g), showing a 25.90% increase over the control (Table 3). Additionally, analysis highlighted a significant impact of stress on RDW, with a 40.10% reduction observed in the control. Among the tested treatments, biopriming with EPSH ITA emerged as the most effective, with a 30.20% increase in RDW compared to the control. Moreover, the research findings unequivocally demonstrated that drought stress conditions significantly impacted SVI, resulting in a 53.40% reduction in SVI in the control group (Table 3). Notably, all treatments investigated exhibited a significant effect on this parameter compared to the control, both under optimal and drought-stressed conditions. Under optimal conditions, the treatment with IPSH SRB showed the highest SVI values compared to the control (1694.60), representing a notable 24.90% increase over the control. Moreover, it is evident that PSH derived from the SRB strain, consistent with previous findings, exerted the most beneficial effects under drought conditions. A significant increase of 100.00% (EPSH SRB) and 109.00% (IPSH SRB) relative to the control was observed, emphasizing their efficacy in enhancing seed vigor even under challenging conditions.

Examining the effects of pea biopriming with PSH includes the monitoring of the SER, RER, R/S ratio, MSI and EL as well (Table 4). The results show, as with the previous parameters, that the drought significantly reduced the rate of SER and RER in the control. Also, the analysis of variance showed a significant increase in SER in the treatments with EPSH ITA and EPSH SRB (11.40% and 17.60%, respectively) compared to the control under optimal conditions. In drought conditions, all tested treatments significantly increased this parameter and among them, the treatments with extracts of the SRB strain stand out the most (54.70% and 50.80%). Under drought conditions, the PSH of the ITA strain also showed a positive effect on SER, but to a lesser extent than the PSH of the SRB strain. Likewise, the results also revealed a significant effect of treatment on the R/S ratio compared to the control. Under optimal conditions, a notable increase in this parameter was observed following biopriming with EPSH SRB (2.03) compared to the control (1.72), while a significant decrease was noted after biopriming with IPSH SRB (1.00). In water deficit conditions, a significant increase was recorded following HP and biopriming (2.65 ± 0.21, EPSH ITA = 2.63 ± 0.48 and EPSH SRB = 2.83 ± 0.09, respectively) with the PSH from the ITA strain.

Conversely, treatments involving PSHs from the SRB strain exhibited a significant decrease in the R/S ratio compared to the control. Furthermore, the obtained results showed a significant variation of the MSI and EL parameters between the examined treatments. The results of the research showed that the drought affected the reduction in the MSI (10.30%) and the increase in EL (43.90%) in the control. Also, a similar but reversed pattern was observed in the response to the investigated treatments. As such, in optimal conditions, it was recorded that biopriming with IPSH ITA had the most significant positive effect on these parameters compared to the control. In conditions of water deficit, the best effect was shown by the PSH of the ITA strain compared to the control (Table 4).

The study also monitors RWC, DTI, SLSI, and RLSI (Table 5). Stress significantly impacted RWC, with a reduction of 11.10% observed in the control group due to drought. The PSH from the SRB strain exhibited the most notable effect on increasing this parameter, with an increase of 3.20% (IPSH SRB) under optimal conditions and 8.10% (IPSH SRB) under drought stress. Moreover, the tested treatments did not significantly affect the DTI under optimal conditions, except for biopriming with EPSH ITA, which showed a significant increase of 13.00% compared to the control. However, under water deficit conditions, all treatments led to a significant increase in the DTI compared to the control. EPSH ITA (23.00%) and EPSH SRB (24.00%) exhibited the most favorable effects among the tested treatments. Regarding the SLSI and RLSI, the treatments had varying effects on these parameters. Under optimal conditions, the HP and ITA strain PSH led to a significant increase in the SLSI compared to the control. In drought conditions, a statistically significant increase in the SLSI was observed after biopriming with ITA and SRB strain PSH. Additionally, all examined treatments except HP had a positive effect on the RLSI under both optimal and drought stress conditions (Table 5).

### 3.3. Correlation Analysis

Correlation analysis confirmed the significant effect of biopriming with PSHs isolated from *S. commune* on seed quality and initial growth of pea under optimal and drought stress conditions (Appendix A). Results indicated that under optimal conditions, there were highly significant positive correlations between various parameters as follows: the SL and RFW, as well as the RLSI; RL and RER, alongside the SVI; SFW and SDW, coupled with the MSI; RFW and RLSI; RDW and RER, alongside EL. Conversely, significant negative correlations were observed between the SFW and EL, RDW and MSI, RER and MSI, and between the MSI and EL (Appendix A).

A strong and significant correlation (r^2^ > 0.5) was identified among various plant parameters and drought tolerance indices. These included relationships between GE and SG, RFW and RDW, SL and RL, SER and RER, SVI, RWC, SLSI, and RLSI, as well as other key factors. Conversely, negative correlations (r^2^ < 0) were observed between the SER and the R/S ratio, as well as MSI and EL under drought stress (Appendix A).

## 4. Discussion

The stages of seed germination and emergence are crucial for achieving an optimal crop composition. Various factors, such as conditions during seed development and subsequent storage, play a significant role in seed germination and vigor, thus greatly impacting crop yields [4]. Moreover, drought stress negatively impacts plant growth and productivity, but seed priming has been shown to mitigate these effects by inducing a primed state in seeds, resulting in early and uniform germination, enhanced growth features, and improved stress response in plants [32]. The effects of drought on plants range from morphological to molecular levels, affecting crop growth, yield, and various physiological processes like cell division and enlargement [33,34]. Therefore, assessing the quality of pea seeds and their capacity to generate healthy shoots under both optimal and drought-stressed conditions is significant.

The observed decrease in GE and SG under drought stress in this study is likely attributed to prolonged imbibition phase III duration and the hindrance of oxygen supply to seeds during germination due to the high viscosity of PEG and limited O_2_ diffusion [23,35,36]. However, the findings reveal that PSH derived from *S. commune*, under submerged cultivation, promotes both the GE and SG of pea seeds. Furthermore, they demonstrate a beneficial effect on the initial plant growth and development, under both examined conditions. These results mark a pioneering contribution, as there is no literature data to the best of our knowledge on the utilization of PSH from this fungus in seed biopriming.

Traditionally, seed biopriming involves the application of various bioinoculants, such as plant extracts, beneficial microorganisms (bacteria, algae, fungi, etc.), or biological products (e.g., chitosan, humic acid), known to enhance specialized characteristics to mitigate the adverse effects of abiotic stress and boost yields [2]. For instance, previous studies have shown that biopriming with cattail extract (*Typha angustifolia*) increased the germination energy of pea seeds by 15.00% under stressful conditions like salinity [37].

In contrast, our research demonstrates an even more significant improvement, with a 31.50% increase in germination energy after biopriming with PSH (IPSH ITA) in drought conditions. Similarly, biopriming with PSH isolated from the microalgae *Chlorella vulgaris* has been shown to enhance seed germination and initial plant growth parameters in wheat and beans [38], while treatment with EPSH derived from rhizobacteria significantly stimulated seed germination, growth, and yield in wheat [39]. On the other hand, priming of garden peas with calcium chloride (osmopriming) and salicylic acid (hormopriming) significantly influenced all analyzed parameters of seed germination and initial plant growth, except for the proportion of abnormal seedlings, both in optimal and stressful conditions (heat stress) [25]. Compared to EPSH and IPSH isolated from the submerged culture of *S. commune*, GE under optimal conditions after hormo- and osmopriming was increased by 0.10 to 7.00%, respectively, on average, i.e., under heat stress conditions by 2.30 to 14.30% [25].

Furthermore, Miljaković et al. [4] showed that biopriming of soybean seeds with bacteria *B. japonicum* and *B. megaterium* significantly affects the increase in seed quality as well as improved plant growth. The greater SL and RL compared to the control may be due to increased divisions in the apical meristem, which caused an increase in initial growth, as is the case after biopriming of pea seeds with cattail extract [37]. Research by Shaffique et al. [40] showed a positive impact of biopriming with a bacterial strain similar to *Klebsiella spp*. which produces an EPSH matrix on seed germination, vigor and wheat biomass production under drought conditions, which is in agreement with the results of this study. Also, biopriming of the pea seeds with *Trichoderma asperellum* affected the increase in the SL up to 22%, RL up to 23%, and the RDW (60%) and SDW (21%) under optimal conditions after 40 days [41]. Furthermore, Chandra Nayaka et al. [42] indicated a positive effect of bioprimed corn seeds with *Trichoderma harzianum* on SG and vigor index, which agrees with the obtained results. Ghezal et al. [37] attributed the higher seed germination rate and germination uniformity to metabolic repair during imbibition, as well as the accumulation of secondary metabolites that increase germination. Namely, seed germination, as an essential phase of the plant growth, development, and successful establishment of crops, is threatened in drought conditions due to weaker activation of the necessary hydrolytic enzymes for starch breakdown, solubilization and transport of carbohydrates, which leads to a decrease in vigor and germination index [43]. These comparisons underscore the remarkable potential of PSHs from *S. commune* in seed biopriming, offering promising prospects for enhancing crop productivity, particularly under challenging environmental conditions.

Given that seed quality affects the speed and uniformity of pea emergence, as well as the initial plant growth [25], quality parameters of initial pea growth after biopriming with fungal PSH in optimal conditions and conditions of water deficit—drought were also examined. The qualitative parameter of SVI was closely monitored, serving as a reliable indicator of seed vitality and robustness. Seed vigor index encompasses a multitude of traits, including tolerance to aging, dormancy, viability, rapid germination, and shoot formation, particularly under adverse conditions. It is influenced by various genetic factors and external environmental influences [44]. Moreover, the R/S ratio serves as a crucial parameter indicating the balance between plant tissue dedicated to support functions (root) and that promoting growth (shoots). This parameter is heavily influenced by nutrient availability and seed mass [45], making its monitoring essential in these studies. The monitoring of MSI and EL was of great importance, considering that these parameters indicate the impact of oxidative stress in drought conditions and damage to lipid membranes due to the creation of reactive oxygen species, especially in photosynthetic organelles [46]. According to Almeselmani et al. [47], this can lead to direct oxidation of lipid membranes so that their permeability increases and causes ion leakage. Moreover, RWC serves as a crucial indicator of plant water status, reflecting the balance between leaf tissue water supply and transpiration rate [48]. The findings of this study revealed that PSH extracted from two strains of *S. commune* positively impacted all these parameters, particularly in drought conditions, further affirming the potential of these metabolites as biopriming agents.

Priming is recognized for its capacity to rectify damage induced by seed aging and exposure to abiotic stresses [37]. The observed biostimulatory effects on seed germination and initial growth parameters attributed to isolated EPSH and IPSH can likely be ascribed to the PSH’ ability to mitigate water loss. Previous studies on cyanobacterial PSH, like those from *Chlorella vulgaris*, have demonstrated their capability to alleviate osmotic disturbances in seeds, ensuring adequate moisture for germination [38,49]. The FTIR spectra of isolated EPSH and IPSH from two tested *S. commune* strains (Figure 1) have coincided with the FTIR spectra of polysaccharide extract samples from our previous research, where we have proven that based on a similar pattern and matching the literature data these polysaccharides are schizophyllan [17]. Moreover, in FTIR spectra from this and our previous study [17] peaks characteristic for the presence of predominantly polysaccharide molecules, a small amount of protein and some aromatics were also observed. Although results of FTIR analysis of EPSH and IPSH of SRB and ITA strain were similar, compared to EPSH sourced from *S. commune* elements in China, which displayed lower proportions of carbon, hydrogen, and nitrogen (C: 25.84%, H: 5.45%, N: 0.65%), the EPSH extracted from the SRB strain exhibited notably higher percentages of all three compounds (C: 37.16%, H: 6.69%, N: 3.16%) [50]. This observation suggests that proteins are likely not completely separated from the protein-glucan complexes to which they are attached, given that PSH-rich samples typically contain minimal nitrogen content, primarily below 1% [51]. Conversely, the determination of protein content in isolated IPSH revealed levels below 1% in both strains suggesting that these are likely water-soluble proteins. This implies that the varying effects of EPSH and IPSH may stem from differences in their structure, largely influenced by the varying ratios of proteins in protein-glucan complexes.

Moreover, considering the positive correlation between GE and initial growth parameters under drought conditions, it is evident that biopriming with PSH from *S. commune* holds critical importance for early-stage development (germination) and subsequent biomass increase and yield enhancement during pea development. This aligns with findings by Saha et al. [43] highlighting the increase in drought tolerance and enhanced seed germination under drought conditions attributed to the growth-promoting effects of biopriming. Furthermore, the presented results underscore the significance of biopriming in maintaining and enhancing physiological parameters like MSI and RWC under water deficit conditions. Drought’s impact on morphological, physiological, biochemical, and molecular characteristics during germination and emergence phases can significantly impede shoot growth [52]. In this context, priming plays a crucial role in improving germination and plant growth by activating numerous stress-responsive genes, regulating proteins and genes involved in various cellular processes (such as drought-responsive RD1 and RD2 genes of the AP2/ERF TF family as well as P5CSA encoding pyrroline-5- carboxylate synthase A, a key enzyme in proline synthesis), and facilitating the mobilization of reserve substances, among other functions [43,53]. For instance, cyanobacterial filtrates have been found to stimulate the synthesis of bioactive compounds, including cytokinins, auxins, and gibberellins, influencing root and shoot growth in germinated wheat seeds [38,54]. Additionally, studies on biopriming of *Trichoderma harzianum* wheat seeds under drought conditions have demonstrated its effectiveness in improving MSI and RWC [55], consistent with the findings of this study. Bouremani et al. [56] also noted a significant reduction in RWC due to drought, leading to protein and enzyme denaturation, membrane instability, and metabolic imbalance in cells. Their research on plant growth-promoting rhizobacteria (PGPR) treatments to mitigate drought effects highlighted favorable impacts on RWC and membrane stability, indicating the need for further investigation at cellular and molecular levels to elucidate underlying mechanisms.

## 5. Conclusions

In conclusion, this preliminary investigation into fortifying *P. sativum* L. seeds in optimal conditions and under drought stress using PSH from *S. commune* sheds light on a promising avenue for sustainable agriculture in the face of escalating environmental challenges. Our preliminary findings underscore the significant potential of IPSH and EPSH, with emphasis on SRB strain as biostimulants for enhancing plant resilience to drought stress.

These results are pioneering, because there is no previous literature data on the use of PSH from this fungal species in the context of seed biopriming. The results showed that PSHs from *S. commune*, cultivated in submerged culture, positively contributes to GE and SG of pea seeds, with an emphasis on PSH isolated from the SRB strain. Likewise, PSH have a beneficial effect on the initial growth and development of plants when exposed to drought stress, since all tested treatments led to a significant increase in the DTI compared to the control, and the best effect was shown by EPSH ITA (23.00%) and EPSH SRB (24.00%). Moreover, the FTIR analysis and microanalysis indicate that proteins are probably not entirely detached from the protein-glucan complexes to which they are linked, which might explain the observed differences in the investigated effects.

Further research is warranted to explore the broader applicability of *S. commune* PSH across different plant species and environmental contexts. Additionally, investigations into the molecular mechanisms underlying the interaction between these PSH and plant physiology will deepen our understanding and facilitate the development of tailored strategies for enhancing crop resilience.

## Figures and Tables

**Figure 1 microorganisms-12-01107-f001:**
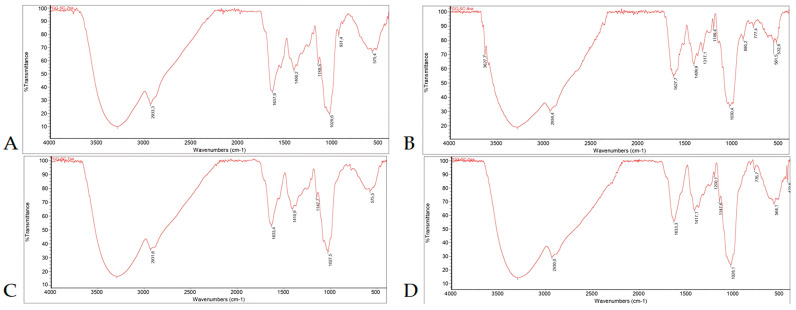
FTIR analysis of the isolated EPSH and IPSH from *S. commune* ((**A**) EPSH SRB; (**B**) IPSH SRB; (**C**) EPSH ITA; (**D**) IPSH ITA).

**Figure 2 microorganisms-12-01107-f002:**
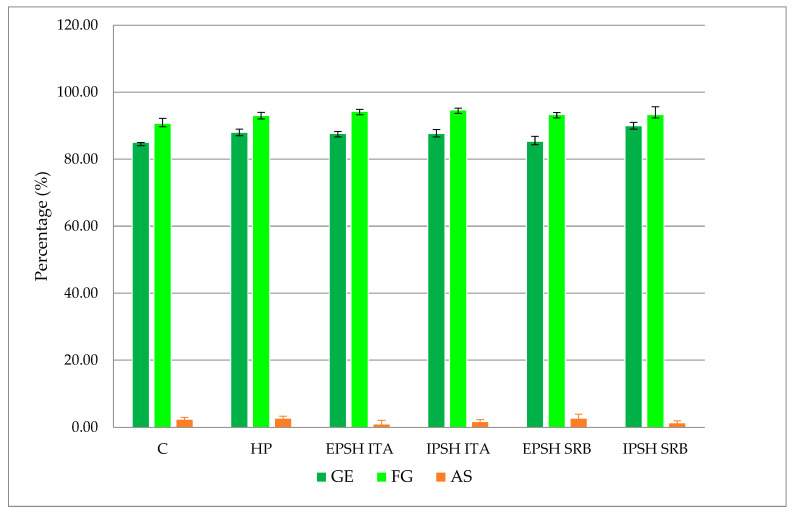
The effect of biopriming with PSHs isolated from both strains *S. commune* (SRB, ITA) under optimal conditions on the parameters of pea seed quality testing (GE—germination energy, FG—final germination, AS—atypical shoots).

**Figure 3 microorganisms-12-01107-f003:**
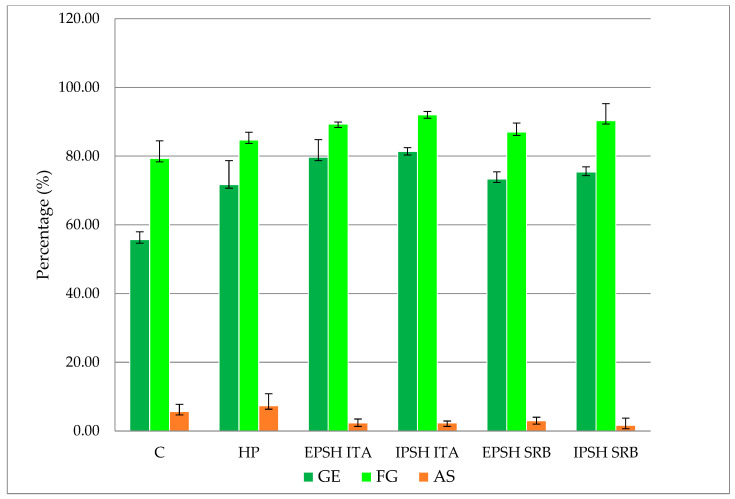
The effect of biopriming with PSHs isolated from *S. commune* under drought conditions on the parameters of pea seed quality testing (GE—germination energy, FG—final germination, AS—atypical shoots).

**Table 1 microorganisms-12-01107-t001:** Results of elemental organic microanalysis and total protein content (TPC).

Content (%)	IPSH SRB	IPSH ITA	EPSH SRB	EPSH ITA
N	3.15	2.75	3.16	2.69
C	38.40	37.53	37.16	36.22
S	nd	nd	nd	nd
H	6.74	6.46	6.69	6.42
TPC (%)	0.9	0.6	na	na
TPC (mg BSA/g d.w.)	175.12 ± 11.25	171.39 ± 4.97	na	na

nd—not detected; na—not analyzed.

**Table 2 microorganisms-12-01107-t002:** Two-factor analysis of variance of quality parameters of pea after biopriming with PSH isolated from *S. commune* under optimal and drought stress conditions.

Trait	S	T	S × T
Germination energy (GE)	***	***	***
Seed Germination (SG)	***	***	***
Abnormal seedlings (AS)	**	**	NS
Shoot length (SL)	***	***	***
Root length (RL)	***	***	***
Shoot fresh weight (SFW)	***	**	***
Root fresh weight (RFW)	***	***	***
Shoot dry weight (SDW)	***	***	***
Root dry weight (RDW)	***	***	***
Shoot elongation rate (SER)	***	***	***
Root elongation rate (RER)	***	***	***
Seedling vigor index (SVI)	***	***	***
Membrane stability index (MSI)	***	***	***
Electrolyte leakage (EL)	***	***	***
Relative water content (RWC)	***	***	**
Drought tolerance index (DTI)	***	***	***
Shoot length tolerance index (SLSI)	***	***	***
Root length tolerance index (RLSI)	***	***	***

** *p* < 0.01; *** *p* < 0.001; NS—not significant.

**Table 3 microorganisms-12-01107-t003:** Effects of biopriming with PSH isolated from *S. commune* on seedling quality parameters (SH, RH, SFW, RFW, SDW, RDW and SVI) in optimal and drought conditions.

Treatment *	SL ** (mm)	RL (mm)	SFW (g)	RFW (g)	SDW (g)	RDW (g)	SVI
Optimal conditions							
C	55.00 ± 1.50 ^d^	94.70 ± 2.90 ^d^	2.37 ± 0.17 ^ab^	2.08 ± 0.04 ^c^	0.2064 ± 0.009 ^bc^	0.1820 ± 0.001 ^c^	1356.80 ± 7.30 ^e^
HP	59.0 ± 0.90 ^bc^	97.0 ± 2.0 ^d^	2.48 ± 0.11 ^b^	2.16 ± 0.02 ^b^	0.2421 ± 0.015 ^ab^	0.1785 ± 0.002 ^c^	1450.80 ± 21.50 ^d^
EPSH ITA	64.50 ± 0.09 ^a^	114.5 ± 3.30 ^b^	2.76 ± 0.11 ^b^	2.45 ± 0.02 ^a^	0.2569 ± 0.023 ^a^	0.1831 ± 0.001 ^c^	168.70 ± 39.70 ^ab^
IPSH ITA	60.80 ± 0.70 ^b^	105.7 ± 3.3 ^c^	2.48 ± 0.06 ^b^	2.37 ± 0.05 ^a^	0.2092 ± 0.005 ^bc^	0.1847 ± 0.001 ^c^	1576.1 ± 30.6 ^c^
EPSH SRB	57.20 ± 0.70 ^cd^	115.90 ± 1.10 ^b^	1.98 ± 0.16 ^a^	2.20 ± 0.01 ^b^	0.1844 ± 0.008 ^c^	0.2020 ± 0.005 ^a^	1615.50 ± 24.00 ^bc^
IPSH SRB	57.60 ± 2.20 ^bcd^	123.90 ± 1.50 ^a^	2.03 ± 0.25 ^a^	2.13 ± 0.02 ^bc^	0.1957 ± 0.021 ^c^	0.1950 ± 0.002 ^b^	1694.60 ± 29.80 ^a^
Drought conditions							
C	26.00 ± 0.50 ^d^	53.50 ± 4.30 ^e^	1.18 ± 0.12 ^c^	1.42 ± 0.03 ^d^	0.1139 ± 0.003 ^c^	0.1091 ± 0.005 ^c^	632.00 ± 73.40 ^d^
HP	27.30 ± 1.30 ^d^	72.50 ± 3.50 ^d^	1.37 ± 0.05 ^bc^	1.61 ± 0.06 ^bc^	0.1288 ± 0.006 ^b^	0.1291 ± 0.002 ^b^	844.70 ± 9.50 ^c^
EPSH ITA	33.90 ± 0.90 ^b^	88.80 ± 4.20 ^c^	1.35 ± 0.07 ^bc^	1.64 ± 0.01 ^b^	0.1315 ± 0.002 ^b^	0.1420 ± 0.006 ^a^	1096.5 ± 9.50 ^c^
IPSH ITA	30.80 ± 1.10 ^c^	87.30 ± 2.90 ^c^	1.31 ± 0.04 ^bc^	1.81 ± 0.04 ^a^	0.1324 ± 0.003 ^b^	0.1360 ± 0.002 ^ab^	1087.40 ± 44.10 ^b^
EPSH SRB	37.70 ± 0.90 ^a^	119.80 ± 1.90 ^a^	1.62 ± 0.10 ^a^	1.56 ± 0.02 ^bc^	0.1434 ± 0.002 ^bc^	0.1325 ± 0.002 ^ab^	1369.80 ± 42.60 ^a^
IPSH SRB	37.30 ± 0.50 ^a^	108.9 ± 2.20 ^b^	1.42 ± 0.09 ^ab^	1.54 ± 0.03 ^c^	0.1329 ± 0.003 ^b^	0.1332 ± 0.006 ^ab^	1320.90 ± 88.90 ^a^

* Data are presented as the mean of three replicates ± standard deviation. Differences between treatments were analyzed using ANOVA, as well as the post-hoc Tukey’s HSD test (*p* < 0.05). Each column (optimal conditions and drought) has a different superscript indicating statistical significance. ** SL—shoot length, RL—root length, SFW—shoot fresh weight, RFW—root fresh weight, SDW—shoot dry weight, RDW—root dry weight, SVI—seedling vigor index.

**Table 4 microorganisms-12-01107-t004:** Effect of biopriming with PSH isolated from *S. commune* on seedling quality parameters (SER, RER, R/S ratio, MSI, and EL) in optimal and drought conditions.

Treatment *	SER **	RER	R/S Ratio	MSI (%)	EL (%)
Optimal conditions					
C	20.21 ± 0.41 ^c^	9.19 ± 1.10 ^b^	1.72 ± 0.10 ^b^	81.00 ± 0.18 ^b^	19.00 ± 0.18 ^b^
HP	10.21 ± 0.22 ^c^	9.14 ± 0.91 ^b^	1.64 ± 0.06 ^b^	81.88 ± 0.52 ^b^	18.12 ± 0.52 ^b^
EPSH ITA	11.37 ± 0.21 ^ab^	10.94 ± 1.22 ^b^	1.78 ± 0.06 ^b^	81.78 ± 0.38 ^b^	18.22 ± 0.38 ^b^
IPSH ITA	10.47 ± 0.37 ^bc^	10.80 ± 1.36 ^b^	1.74 ± 0.05 ^b^	83.99 ± 0.22 ^a^	16.01 ± 0.22 ^c^
EPSH SRB	12.01 ± 0.39 ^a^	14.78 ± 0.47 ^a^	2.03 ± 0.04 ^a^	71.07 ± 0.51 ^c^	28.93 ± 0.51 ^a^
IPSH SRB	10.07 ± 0.62 ^c^	13.92 ± 0.65 ^a^	1.00 ± 0.07 ^c^	71.67 ± 1.06 ^c^	28.33 ± 1.06 ^a^
Drought conditions					
C	5.91 ± 0.30 ^c^	5.73 ± 1.50 ^d^	2.05 ± 0.38 ^b^	72.66 ± 1.04 ^b^	27.34 ± 1.04 ^a^
HP	4.40 ± 0.41 ^d^	6.83 ± 1.25 ^d^	2.65 ± 0.21 ^a^	72.48 ± 0.79 ^b^	27.52 ± 0.79 ^a^
EPSH ITA	7.03 ± 0.36 ^b^	13.31 ± 1.73 ^c^	2.63 ± 0.48 ^a^	81.99 ± 0.93 ^a^	18.01 ± 0.93 ^b^
IPSH ITA	6.09 ± 0.35 ^c^	11.50 ± 1.17 ^c^	2.83 ± 0.09 ^a^	82.37 ± 0.81 ^a^	17.63 ± 0.81 ^b^
EPSH SRB	9.14 ± 0.28 ^a^	22.23 ± 1.69 ^a^	0.92 ± 0.02 ^c^	71.93 ± 0.67 ^b^	28.07 ± 0.67 ^a^
IPSH SRB	8.91 ± 0.25 ^a^	18.23 ± 0.58 ^b^	1.00 ± 0.11 ^c^	72.15 ± 1.04 ^b^	27.85 ± 1.04 ^a^

* Data are presented as the mean of three replicates ± standard deviation. Differences between treatments were analyzed using ANOVA, as well as the post hoc Tukey’s HSD test (*p* < 0.05). Each column (optimal conditions and drought) has a different superscript indicating statistical significance. ** SER—shoot elongation rate, RER—root elongation rate, R/S ratio—root/shoot ratio, MSI—membrane stability index, EL—electrolyte leakage.

**Table 5 microorganisms-12-01107-t005:** Effects of biopriming with PSHs isolated from *S. commune* on seedling quality parameters (RWC, DTI, SLSI and RLSI) in optimal and drought conditions.

Treatment *	RWC **	DTI	SLSI	RLSI
Optimal condition				
C	81.80 ± 0.70 ^bc^	1.00 ± 0.00 ^b^	100.00 ± 0.00 ^c^	100.00 ± 0.00 ^d^
HP	81.90 ± 0.30 ^bc^	1.08 ± 0.03 ^ab^	107.30 ± 1.80 ^b^	102.50 ± 1.70 ^cd^
EPSH ITA	81.30 ± 1.20 ^c^	1.13 ± 0.06 ^a^	117.30 ± 2.00 ^a^	121.20 ± 6.70 ^ab^
IPSH ITA	82.30 ± 0.50 ^bc^	1.01 ± 0.03 ^b^	110.70 ± 3.80 ^ab^	111.70 ± 4.40 ^bc^
EPSH SRB	83.10 ± 1.10 ^ab^	0.99 ± 0.02 ^b^	104.00 ± 3.40 ^bc^	122.60 ± 4.60 ^ab^
IPSH SRB	84.40 ± 0.50 ^a^	1.01 ± 0.04 ^b^	104.70 ± 2.10 ^bc^	131.00 ± 3.20 ^a^
Drought conditions				
C	72.70 ± 1.10 ^bc^	1.00 ± 0.00 ^c^	100.00 ± 0.00 ^d^	100.00 ± 0.00 ^d^
HP	71.80 ± 0.60 ^c^	1.16 ± 0.03 ^b^	105.20 ± 6.00 ^d^	135.90 ± 8.80 ^cd^
EPSH ITA	75.30 ± 0.20 ^ab^	1.23 ± 0.01 ^a^	130.40 ± 0.80 ^b^	167.20 ± 22.20 ^bc^
IPSH ITA	74.20 ± 0.8 ^bc^	1.20 ± 0.01 ^ab^	118.60 ± 5.00 ^c^	164.20 ± 18.10 ^bc^
EPSH SRB	75.80 ± 1.90 ^ab^	1.24 ± 0.01 ^a^	144.90 ± 2.60 ^a^	224.80 ± 15.40 ^a^
IPSH SRB	78.60 ± 1.80 ^a^	1.19 ± 0.03 ^ab^	143.40 ± 4.30 ^a^	204.50 ± 18.06 ^ab^

* Data are presented as the mean of three replicates ± standard deviation. Differences between treatments were analyzed using ANOVA, as well as the post hoc Tukey’s HSD test (*p* < 0.05). Each column (optimal conditions and drought) has a different superscript indicating statistical significance. ** RWC—relative water content, DTI—drought tolerance index, SLSI—shoot length tolerance index, RLSI—root length tolerance index.

## Data Availability

The data presented in this study are available on request from the corresponding author. The data are not publicly available due to privacy.

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
