# Peer review of "Unveiling Fungi Armor: Preliminary Study on Fortifying Pisum sativum L. Seeds against Drought with Schizophyllum commune Fries 1815 Polysaccharide Fractions"

_microorganisms, 2024, doi:10.3390/microorganisms12061107_

Round 1

Reviewer 1 Report

Comments and Suggestions for Authors

The content of the results of this manuscript is not the focus of the journal

Other suggestions and comments are included in the manuscript.

Authors must demonstrate the following:

1) the chemical characterization of the supposed bioactive fraction (poly-, oligo-, disaccharides or monosaccharides) with a concentration-response curve.

2) that chemical and physical characterization of the supposed bioactive fraction is unnecessary.

3) That the biological variability of Pisum sativum seed and Schizuphyllum commune is an inconsequential factor in the fortifying effect of fungal polysaccharides

4) congruence between the title of the manuscript and the purpose of the research; the conclusion of the abstract section and the conclusion of the discussion section.

5) That controls are not necessary, e.g. different varieties of Pisum sativum and include several isolates of Schizuphyllum commune and the inclusion of “biologically inert” polysaccharides

6) That it is necessary to include acronyms in the abstract

Comments on the Quality of English Language

no comments

Reviewer 2 Report

Comments and Suggestions for Authors

Dear authors,

I have read your manuscript with interest. Currently, the search for new ways to improve crop growth is among the interests of many researchers.  The range of substances and agronomic techniques offered is very wide. The authors of the article add to it another substance that has not previously been studied as a treatment for seeds. The idea of the study is interesting and original. The design of the study is mostly correct and allows you to draw reasonable conclusions. However, I have comments on the text of the article and especially on the interpretation of the results.

1.   The authors extrapolate the results of one laboratory experiment with one plant species to the cultivation of agricultural plants as a whole. Line 505 «Moreover, considering the positive correlation between GE and initial growth parameters under drought conditions, it is evident that biopriming with PSH from S. commune holds critical importance for early-stage development (germination) and subsequent biomass increase and yield enhancement during pea development». Line 546 «Significant improvement of the tested parameters, as well as a positive correlation  between GE and initial growth parameters under drought conditions indicate that the application of these PSH may have long-term benefits for the plant». The authors observed only the phase of seed germination and initial growth (8 days). The article does not substantiate or prove the long-term effect of seed treatment with polysaccharides. I believe that all unfounded assumptions should be removed from the text of the article. On the contrary, it should be indicated whether protection from water deficiency in the seed germination phase is of independent importance. Or is the presented work only the first stage of the research, and the significance of the research can be assessed only after the end of the field experiments.

2.   It is unclear why the authors tested both exo-polysaccharides and intra-polysaccharides. How do these two groups of polysaccharides differ in composition and properties? Exo- and intra-polysaccharides had different effects on some indicators of pea seedlings. Is this due to differences in the composition of exo- and intra-polysaccharides?

3.   The materials and methods describe a method for producing polysaccharides, but do not describe the properties of the finished product. This makes it difficult to compare the results of this study and similar scientific publications.

4.   Why were only peas used as test plants? Perhaps the observed effect is specific to peas and is not reproduced for other crops. Would the authors like to repeat the experiments with other plant species to check the reproducibility of the effects before publishing the results?

5. Short note. The AS columns are poorly visible in Figures 1 and 2. 

Round 2

Reviewer 1 Report

Comments and Suggestions for Authors

1)     An apology to the authors of the manuscript microorganisms-2991774 for pointing out that "this manuscript is not the focus of the journal". My argument is that the invitation to review the manuscript did not indicate the focus of the Special Issue. Once again an apology.

2)     Answer 1.10 is unacceptable. The argument of testing time with adequate controls for obtaining data is very weak, unrelated to the purpose of publishing the manuscript. On the other hand, teaching a Nobel doctor that preliminary data without adequate controls are publishable in the classic publication format is not the best training.

3)     Opinion

However, the title, purpose, introduction and conclusion must be consistent in the context of the study: preliminary observations. Please make the title, purpose, introduction and conclusion consistent.

4) Comments on the manuscript

Comments on the Quality of English Language

No comment

Author Response

Reviewer #1

Authors' response: Dear Reviewer #1, we have carefully addressed all the comments and suggestions provided and believe that the revised manuscript now meets the standards of rigor and quality expected by Microorganisms journal. Changes have been marked in Microsoft Word using the Track Changes feature and highlighted in blue, for the second round of revision (highlights in yellow are from the first round of revision). A comprehensive response to all Your remarks is presented in blue below, with references to page numbers and lines in the revised manuscript indicated in red.

  1. Reviewer #1 has expressed regret for the oversight in the first revision, acknowledging that the content of our manuscript's results does not align with the focus of the journal. They argue that the invitation to review the manuscript did not clearly indicate the focus of the Special Issue, and they extend their apologies once again.

Authors' response: Dear Reviewer #1, thank You for your understanding and gracious acknowledgement. Your apology is genuinely appreciated, and we want to assure you that all is well. Your insights have been invaluable in guiding us through the revision process, and we are grateful for your time and effort in providing constructive feedback.

  1. Reviewer #1 deemed our response to question 1.10 unacceptable. Reviewer #1 criticized our argument regarding testing time and adequate controls for data acquisition as weak and irrelevant to the manuscript's purpose for publication. Additionally, Reviewer #1 expressed concern that advocating for the publication of preliminary data without proper controls may not provide the best training, even for a Nobel laureate.

Authors' response: Dear Reviewer #1, we believe that your complaint in terms of control is completely scientifically justified, however the main goal of this study was to use naturally available biological sources of polysaccharides that are always in complexes (for fungi it is known that polysaccharide protein complexes dominate) which acts synergistically but not isolated compounds with a chemically defined structure (e.g. standard schizophyllan in this case), therefore we cannot chemically confirm the exact structure of the obtained mixture from the obtained fraction. The only confirmation of the structure in this work was obtained through FTIR analysis, which certainly confirms that it is a mixture dominated by beta-glucan with admixtures of proteins and aromatic compounds (phenols in the case of mushrooms). Based on that, it follows that the control with schizophyllan would not be adequate enough when it comes to the potential control of our fraction, because then we would have to examine some peptides and phenolics as additional controls. Based on the results of our previous research with S. commune it was known that the fractions are complex mixtures, and our goal was directed towards a new way of organic biopriming as the cheapest and most natural approach to organic agriculture.

Given that the goal of this study was to examine the potential of polysaccharide fractions unexplored so far (added now at the title of the revised MS) that originates from S. commune, while not the analysis of schizophyllan present as the probably dominant component of the fraction, we considered that it was not necessary to control the pure compound, i.e. chemically defined controls. Nonetheless, we appreciate Your valuable suggestions provided, which will undoubtedly enhance future research endeavors.

Furthermore, since this research marks pioneering strides in exploring novel biostimulants derived from macrofungi for seed priming, adequate controls were implemented following established protocols in seed science. Thus, the control groups, comprising non-primed seeds and hydroprimed seeds utilized to eliminate water's effects, were deemed appropriate for this study, given that priming entails hydrating seeds to activate their metabolism. Furthermore, it's essential to highlight the reason behind utilizing hydropriming (HP) as a secondary control, as the polysaccharides (IPSH and EPSH) extracted from S. commune were in the form of aqueous solutions, in which seed were bioprimed. This choice ensures methodological rigor during this phase of the research.

Reference: Mišković, J.; Karaman, M.; Rašeta, M.; Krsmanović, N.; Berežni, S.; Jakovljević, D.; Piattoni, F.; Zambonelli, A.; Gargano, M.L.; Venturella, G. Comparison of two Schizophyllum commune strains in production of acetylcholinesterase inhibitors and antioxidants from submerged cultivation. J. Fungi 2021, 7, 115. https://doi.org/10.3390/jof7020115 

  1. Reviewer #1 emphasized the importance of ensuring consistency between the title, purpose, introduction, and conclusion, particularly in the context of preliminary observations. Reviewer #1 requested that we align these elements to maintain coherence throughout the manuscript.

Author’s response:  Thank you for your insightful comments regarding the coherence and consistency of our manuscript. We have carefully considered your feedback and are committed to aligning the title, purpose, introduction, and conclusion to ensure a seamless flow of ideas throughout the paper.

To address this concern, we have revised the title to accurately reflect the core objectives of our preliminary study (page 1, line 2-3), ensuring it is in harmony with the purpose stated in the Abstract (page 1, lines 23-24 and 38) and Introduction (page 2, line 93 and page 3, lines 107, 109-111). Furthermore, we recognize the importance of integrating preliminary observations cohesively into the conclusion to reinforce the key findings and their implications(page 17, lines 615 and 618). Hence, we have ensured that the conclusion serves as a logical endpoint that resonates with the initial objectives outlined in the introduction. Overall, we appreciate your guidance in enhancing the clarity and coherence of our manuscript. Your suggestions will undoubtedly strengthen the overall impact of our research.

  1. 4. Reviewer #1 provides additional comments on the manuscript.

4.1.Page 4, line 174: Reviewer #1 commented: “we chose a drought threshold (-0.49 MPa) that considerably lowers the initial growth of shoots, as we have shown in previous research. In these settings, pea seedlings were drought tolerant for three weeks, after which the laboratory experiment was concluded, despite significantly lower growth and biomass accumulation in the control.”

Author’s response: Dear Reviewer #1, the following was added in2. Materials and Methods, subsection 2.5.1. Seed Germination Assessment(page 4, line 194 – 196):

The experiment consisted of a total of 36 boxes, grouped into two sets, where one set contained 18 boxes that were optimally supplied with water, and the other contained 18 boxes in which drought was simulated using a solution of polyethylene glycol-0.5 MPa solution of polyethylene glycol (PEG-6000) (Sigma Aldrich, St. Louis, MO, USA), which was proved to be a drought threshold (medium stress) that significantly reduce pea seedling growth, as outlined by Tamindžić et al. [23].

4.2.Page 6, line 324: Reviewer #1 commented: “Please improve the IR-spectra to eliminate the excess humidity observed in the region from 3700 to 2300 cm-1.”

Author’s response: Dear Reviewer #1, following your recommendation, Dr. Gordana Gojgić-Cvijović, principal research fellow, has conducted further analysis on the FTIR spectra. As a result, we have incorporated additional spectra into our revised manuscript (page 7). The FTIR spectra of four samples (      EPSH and IPSH for both SRB and ITA strains) were recorded after the samples were dried overnight in a vacuum oven at +40°C. Upon processing the spectra, no significant differences were observed; however, some of the peaks appeared to be better expressed. I hope the new spectra meet Your criteria. Below, You will find images of all the individual spectra which were replaces with the previously presented ones.

Figure 1. FTIR analysis of isolated EPSH and IPSH from S. commune

A - EPSH SRB

B - IPSH SRB

C - EPSH ITA

D – IPSH ITA

Reviewer 2 Report

Comments and Suggestions for Authors

Dear authors,

thank you for the detailed answers to the questions. The information you provided made some aspects of the experiment more understandable. I am satisfied with the changes made to the manuscript.The only remark. Why is the result of the EPSH ITA elemental analysis not written? There is no explanation for this in the text. 

I wish you to continue your research successfully.

Author Response

Reviewer #2

  1. Reviewer #2 stated the following: “Dear authors, thank you for the detailed answers to the questions. The information you provided made some aspects of the experiment more understandable. I am satisfied with the changes made to the manuscript. The only remark. Why is the result of the EPSH ITA elemental analysis not written? There is no explanation for this in the text. I wish you to continue your research successfully.”

Authors' response: Dear Reviewer #2, we would like to express our sincere gratitude for the time and effort you invested in thoroughly reviewing our manuscript. Your insightful feedback is valuable to us and has greatly contributed to the enhancement of our paper's quality. Thank You once again for your dedication and constructive criticism in the revision process of our paper.

Furthermore, thank You for noticing this oversight. As requested, we have added the results of elemental analysis for EPSH ITA sample in the Table 1 (page 7, lines 291-301), under the subsection 3.1. FTIR analysis, microanalysis and quantification of total protein content (Section 3. Results). Changes have been marked in Microsoft Word using the Track Changes feature and highlighted in blue, for the second round of revision (highlights in yellow are from the first round of revision):

Table 1. Results of elemental organic microanalysis and total protein content (TPC)

Content (%)

IPSH SRB

IPSH ITA

EPSH SRB

EPSH ITA

N

3.15

2.75

3.16

2.69

C

38.40

37.53

37.16

36.22

S

nd

nd

nd

nd

H

6.74

6.46

6.69

6.42

TPC (%)

0.9

0.6

na

na

TPC (mg BSA/g d.w.)

175.12 ± 11.25

171.39 ± 4.97

na

na

nd – not detected; na – not analyzed.
